# Released ATP Mediates Spermatozoa Chemotaxis Promoted by Uterus-Derived Factor (UDF) in *Ascaris* *suum*

**DOI:** 10.3390/ijms23074069

**Published:** 2022-04-06

**Authors:** Qiushi Wang, Ruijun He, Qi Zhang, Jin Shan, Yanmei Zhao, Xia Wang

**Affiliations:** 1Key Laboratory of RNA Biology, CAS Center for Excellence in Biomacromolecules, Institute of Biophysics, Chinese Academy of Sciences, Beijing 100101, China; qiushiwang@ibp.ac.cn (Q.W.); hrj@ibp.ac.cn (R.H.); 2University of Chinese Academy of Sciences, Beijing 100049, China; igene@outlook.com (Q.Z.); nevermorejin@ibp.ac.cn (J.S.); 3National Laboratory of Biomacromolecules, CAS Center for Excellence in Biomacromolecules, Institute of Biophysics, Chinese Academy of Sciences, Beijing 100101, China

**Keywords:** uterus-derived factor (UDF), sperm chemotaxis, protein serine phosphorylation, adenosine-5′-triphosphate (ATP)

## Abstract

Fertilization requires sperm migration toward oocytes and subsequent fusion. Sperm chemotaxis, a process in which motile sperm are attracted by factors released from oocytes or associated structures, plays a key role in sperm migration to oocytes. Here, we studied sperm chemotaxis in the nematode *Ascaris suum*. Our data show that uterus-derived factor (UDF), the protein fraction of uterine extracts, can attract spermatozoa. UDF is heat resistant, but its activity is attenuated by certain proteinases. UDF binds to the surface of spermatozoa but not spermatids, and this process is mediated by membranous organelles that fuse with the plasma membrane. UDF induces spermatozoa to release ATP from intracellular storage sites to the extracellular milieu, and extracellular ATP modulates sperm chemotaxis. Moreover, UDF increases protein serine phosphorylation (pS) levels in sperm, which facilitates sperm chemotaxis. Taken together, we revealed that both extracellular ATP and intracellular pS signaling are involved in *Ascaris* sperm chemotaxis. Our data provide insights into the mechanism of sperm chemotaxis in *Ascaris suum*.

## 1. Introduction

The mission of male sperm is to fuse with female oocytes and complete fertilization successfully. To fertilize oocytes successfully, spermatids are activated into motile spermatozoa and then attracted by chemoattractants released from oocytes or associated structures. This process is called sperm chemotaxis and promotes successful fertilization [1,2]. Chemoattractants, such as peptides, proteins, hormones and small molecules, are able to attract spermatozoa in various species [3]. Resact and speract are characterized chemoattractants of sperm in marine organisms [4,5]. Allurin, a 21 kDa protein that is released from the outermost jelly layer of *Xenopus* eggs, attracts spermatozoa of frogs and mice [6,7]. The hormones prostaglandins and progesterone have been identified as chemoattractants of mammalian sperm [8,9]. F-series prostaglandins synthesized by oocytes from polyunsaturated fatty acid (PUFA) precursors guide *Caenorhabditis elegans* sperm to spermatheca [10,11]. These chemoattractants induce spermatozoa to regulate their flagellar beating, motility direction, and size and shape during migration to oocytes. The signal cascade of sperm chemotaxis has been studied in a few species. Resact is released from eggs and forms a gradient in seawater. *A. punctulata* sperm bear a surface guanylyl cyclase (GC) that binds resact and directs the sperm to follow this gradient to the egg [12]. Chemoattractants initiate a chemotaxis signal cascade by interacting with their receptors. Sperm induced by chemoattractants evoke changes in intracellular pH, Ca^2+^, cAMP and cGMP and the membrane potential [13,14,15,16,17]. Progesterone activates Ca^2+^ influx through CatSper, which is a Ca^2+^ channel [18]. In the ascidian, *Ciona intestinalis*, calaxin is an axonemal Ca^2+^-binding protein and plays a critical role in sperm chemotaxis promoted by SAAF (sperm activating and attracting factor) [19]. Ca^2+^ contributes to sperm chemotaxis by regulating flagellar beating [20]. Another messenger, cyclic guanosine monophosphate (cGMP), also contributes to *A. punctulata* sperm chemotaxis [21]. Although the signaling pathway of sperm chemotaxis has been studied extensively in sea urchins, the molecular mechanisms underlying sperm chemotaxis in nematodes and mammals still remain enigmatic.

Most mammalian cells release ATP through a regulated process [22,23]. Released ATP, as a signaling molecule, regulates various physiological functions, including chemotaxis in some species [24,25,26]. Released ATP guides neutrophil chemotaxis promoted by fMLP (N-formyl-methionyl-leucylphenylalanine). Purinergic receptors P2Y and A3 mediate neutrophil chemotaxis [27,28]. The ATP receptor P2Y_11_ mediates ATP-enhanced neutrophil chemotaxis in rats [29]. Here, we report that in the nematode *Ascaris suum*, UDF, the protein fraction of the uterus extract, attracts spermatozoa. UDF induces ATP release from spermatozoa, and the released ATP is essential for sperm chemotaxis. Moreover, the serine phosphorylation (pS) level in sperm is elevated by UDF. The increase in pS level facilitates UDF-induced sperm chemotaxis. These results provide insights into the mechanisms of sperm chemotaxis in the nematode *Ascaris suum* and imply that a similar mechanism may exist in other animal species.

## 2. Results

### 2.1. Characterization of Uterus-Derived Factor (UDF)

For successful fertilization, sperm are attracted by chemoattractants released from oocytes or associated structures [5,21]. To test whether components in oocytes or the uterus of *Ascaris suum* attract spermatozoa, oocyte extract (OE) and uterus extract (UE) were isolated. In the double chamber assay, UE attracted more spermatozoa than OE to pass through the polycarbonate membrane containing 8 µm diameter pores when UE and OE had the same protein concentrations of spermatozoa measuring approximately 15 µm, which suggests that spermatozoa adjust their shape to respond to UE. Since the activity of UE was significantly stronger than that of OE (Figure 1A), we thus focus our further analysis on UE.

To further elucidate the characteristics of the components that attracted spermatozoa in UE, UE was heated at 100 °C for 10 min. The activity of heated UE was similar to that of fresh UE, which indicates that the active component in UE was heat resistant (Figure 1B). Next, we tested whether the active component in the UE was a protein. UE was incubated with proteinases, such as trypsin, Pronase and Proteinase K. These proteases worked well, and almost all of the proteins were digested when UE was incubated with Pronase (2 µg/µL), trypsin (2 µg/µL) at 37 °C for 1 h and Proteinase K (1 µg/µL) at 65 °C for 1 h (Appendix A). Accordingly, UE digested by proteases almost completely lost the activity of attracting spermatozoa in the double chamber assay, suggesting that the active component in UE is protein or protein-binding fraction (Figure 1C).

For further purification, (NH_4_)_2_SO_4_ was used to collect proteins from UE. A pellet of 100% (NH_4_)_2_SO_4_ precipitate was suspended in HKB buffer and was able to attract spermatozoa in a dose-dependent manner (Figure 1D). This fraction was named uterus-derived factor (UDF). Next, we performed a protein separation and purification strategy to identify the sperm-attracting protein(s) in UDF (see Section 4 for details). Heparin columns, Q columns and Superdex 200 columns were used successively to separate the active fractions. However, more than 10 proteins were detected in our final active fractions (Appendix A) and we failed to identify potential candidate proteins.

### 2.2. UDF Binds Spermatozoa but Not Spermatids

Chemoattractants interact with receptors on the plasma membrane of spermatozoa and initiate a downstream signaling cascade [30]. To test whether the active component in UDF binds spermatozoa, the final purification fraction (Superdex 200 column fraction) of UDF was labeled with Alexa Fluor 555 dye (hereafter referred to as Alexa 555-UDF) and then incubated with spermatozoa and spermatids. The results show that Alexa555-UDF bound only to the surface of spermatozoa (Figure 2B) and not spermatids (Figure 2A).

To further quantify the proportions of spermatozoa and spermatids labeled by Alexa 555-UDF, we detected these samples with flow cytometry (Figure 2C,D). Based on this assay, we found that 99% of spermatozoa were labeled by Alexa 555-UDF, but only 12% of spermatids were labeled by Alexa 555-UDF (Figure 2C,D). These spermatids were broken, which led to Alexa 555-UDF entering them directly. These data indicate that UDF binds only spermatozoa and not spermatids. This is consistent with the previous finding that sperm chemoattractants only attract activated sperm.

Next, we studied the reason why UDF only binds spermatozoa but not spermatids. One difference between spermatozoa and spermatids is the occurrence of exocytosis in spermatozoa. During nematode sperm activation, in which spermatids are transformed into fertilization-competent spermatozoa, membranous organelles (MOs), which are sperm-specific organelles, fuse with the plasma membrane. In this process, some components in MOs are released to the extracellular milieu, and some membranes and proteins of MOs remain present on the plasma membrane [31]. Therefore, we hypothesized that the receptor of UDF is present in MOs in spermatids and translocates to the plasma membrane via exocytosis of MOs during sperm activation. To test this hypothesis, we used two reagents that affect MO fusion and pseudopodia extension. One is N-phenylanthranilic acid (DPC), an inhibitor of cystic fibrosis transmembrane conductance regulator (CFTR) [32], which inhibits sperm MO fusion but not pseudopodia extension (Appendix A). The other is o-phenanthroline (PHE), a metalloprotease inhibitor and chelator of Zn^2+^, which blocks pseudopodia extension but not MO fusion (Appendix A). The membrane dye FM 1-43 is a marker of MO fusion and only labels the plasma membrane in spermatids (Appendix A). When MOs fuse with the plasma membrane, FM 1-43 enters the MOs via fusion pores and forms puncta beneath the plasma membrane (Appendix A). In the UDF binding assay, Alexa 555-UDF bound to the surface of PHE-treated sperm (Appendix A) but not DPC-treated sperm (Appendix A). Spermatids and spermatozoa were used as negative and positive controls, respectively (Appendix A). These results indicate that MO fusion is critical for UDF binding on the surface of spermatozoa and provide evidence to support the hypothesis that the receptor of UDF is translocated to the plasma membrane of spermatozoa via MO fusion with the plasma membrane.

### 2.3. UDF Induces ATP to Release from Spermatozoa and Released ATP Is Essential for Sperm Chemotaxis Promoted by UDF

To elucidate the mechanism of sperm chemotaxis induced by UDF, we detected some changes in spermatozoa after UDF induction at the cellular and molecular levels. Intriguingly, we found that UDF enhanced ATP release from spermatozoa. Most mammalian cells release ATP in a regulated manner. Glutamate, acetylcholine, KCl and ATP itself induce ATP release from different cell types [33,34,35]. Quinacrine, an ATP storage marker, labeled spermatozoa and we found that ATP was stored in refringent granules (RGs) which were specific organelles in Ascaris sperm. ATP in RGs was attenuated by UDF, which was confirmed by confocal microscopy imaging (Figure 3A) and flow cytometry analysis (Figure 3B,C). ATP stored in RGs is probably released to the cytoplasm of sperm and even to the extracellular space in response to UDF.

To test this hypothesis, ATP concentrations in the cytosol and extracellular medium were measured with the luciferin–luciferase assay. As shown in Figure 3D,E, UDF induced ATP levels to increase in the cytosol and extracellular milieu in a dose-dependent manner. In addition, the mitochondrial membrane potential of the sperm remained unchanged when spermatozoa were induced by UDF (Figure 3F). This suggests that UDF does not affect mitochondrial activity and does not facilitate ATP synthesis. These results indicate that upon UDF induction, ATP stored in RGs is mobilized to the cytoplasm and then released to the extracellular milieu.

To test the role of extracellular ATP in spermatozoa chemotaxis induced by UDF, apyrase, an ATPase/ADPase, was used to degrade the extracellular ATP. The effect of apyrase (5 U/mL) on spermatozoa chemotaxis was tested. Treatment with apyrase (5 U/mL) attenuated sperm chemotaxis induced by UDF in a dose-dependent manner, indicating the important role of extracellular ATP in sperm chemotaxis (Figure 3G). Moreover, it has been reported that ATP is not a chemoattractant, but it enhances macrophage chemotaxis [36]. We found that adenosine 5′-O (-3-thiotriphosphate) (ATP-γ-S), a nonhydrolyzable ATP analog, facilitated spermatozoa chemotaxis promoted by UDF, which indicates that ATP probably functions as a signaling molecule to modulate sperm chemotaxis (Figure 3H). Our data indicate that extracellular ATP is essential for sperm chemotaxis promoted by UDF.

### 2.4. UDF Elevates Protein Serine Phosphorylation Level of Spermatozoa

Next, we detected some signaling events in sperm induced by UDF and found that phosphoserine (pS) levels in spermatozoa increased after UDF induction (Figure 4A, compare lanes 4 and 1). To further elucidate the functions of pS in sperm chemotaxis, we used inhibitors of serine phosphatase to elevate pS levels in sperm and detected the effects of these inhibitors on sperm chemotaxis induced by UDF. Calyculin A (CA) is an inhibitor of protein phosphatase 2A (PP2A), a Ser/Thr phosphatase, and PP2B. PP2A is present in *Ascaris suum* sperm [37]. Both 10 nM and 100 nM CA increased pS levels in the sperm (Figure 4A). Okadaic acid (OA) is a PP2A-specific inhibitor. Both CA and OA facilitated UDF-induced sperm chemotaxis, suggesting that higher pS levels are beneficial to sperm chemotaxis (Figure 4B,C). Sodium orthovanadate (OV), an inhibitor of tyrosine phosphatases, alkaline phosphatases, and ATPases, elevated tyrosine phosphorylation (pY) levels in spermatozoa. However, the pY level was unaffected by UDF (Appendix A), indicating that pS, but not pY, is involved in UDF-induced sperm chemotaxis.

The relationship between extracellular ATP and protein phosphorylation has been studied in some species. Extracellular ATP stimulates MAP kinase cascades via P2Y receptors in the human Caco-2 cell line [38]. Protein kinase A (PKA) activity is involved in extracellular adenine nucleotide-mediated microglial chemotaxis via P2Y_12/13_ receptors [39]. The PKA pathway is also involved in sperm chemotaxis promoted by progesterone [40]. To test whether PKA mediates spermatozoa chemotaxis promoted by UDF, H89, an inhibitor of PKA, was used to block PKA activity in a sperm chemotaxis assay. H89 attenuated sperm chemotaxis (Figure 4D). This indicates that PKA is required in spermatozoa chemotaxis promoted by UDF. PKA phosphorylates its substrates on serine residues and regulates downstream signaling cascades [41]. Taken together, we propose that UDF induces sperm to release ATP from spermatozoa. Extracellular ATP increases protein serine phosphorylation levels by activating PKA through unidentified ATP receptors localized on the plasma membrane. This signaling cascade is triggered by UDF.

## 3. Discussion

Our findings reveal that in *Ascaris suum*, UDF attracts spermatozoa through active components in UDF that have not been identified. Spermatozoa induced by UDF release ATP to the extracellular milieu (Figure 3E), but UDF did not affect the mitochondrial membrane potential of spermatozoa (Figure 3F). This means that UDF does not affect ATP production. Therefore, UDF only induces ATP release from ATP storage organelles in spermatozoa to the extracellular space.

Our data also show that extracellular ATP is required for nematode sperm chemotaxis. In addition, extracellular ATP is essential for physiological cell functions in some species. Extracellular ATP induces a rapid calcium signal mediated by ATP receptors in mouse caput epididymides [42]. Extracellular ATP induces the acrosome reaction and elevates Ca^2+^ and cAMP levels via ATP receptors in bovine and human spermatozoa [43,44]. The functions of extracellular ATP in chemotaxis were also studied. ATP enhances the chemotaxis of rat neutrophils [29]. In cultured microglia, extracellular ATP enhances chemotaxis via Gi/o-coupled P2Y receptors [45]. However, whether extracellular ATP plays a role in spermatozoa chemotaxis has yet to be studied. Our study demonstrated that extracellular ATP is essential for nematode spermatozoa chemotaxis promoted by UDF. Moreover, we discovered that a nonhydrolyzable ATP analog, ATP-γ-S, facilitated sperm chemotaxis promoted by UDF (Figure 3H). The concentrations of ATP-γ-S we used were higher than extracellular ATP concentrations after UDF treatment (Figure 3E). One possibility is that the products of ATP hydrolysis, ADP, AMP, and adenosine, may also facilitate sperm chemotaxis promoted by UDF via an unknown mechanism. Thus, the nonhydrolyzable ATP-γ-S was less potent than ATP and a higher concentration of ATP-γ-S was required to facilitate sperm chemotaxis.

As mentioned above, the protein serine phosphorylation (pS) level of spermatozoa was also elevated by UDF in addition to extracellular ATP. Increased serine phosphorylation favored spermatozoa chemotaxis induced by UDF. In addition, PKA, a serine kinase, was required for chemotaxis promoted by UDF (Figure 4D). Temporal and spatial regulation of PKA activity is essential for vigorous sperm motility [41]. Serine kinase phosphorylates MSP (major sperm protein) fiber protein 2 (MFP2) to focus on polymerizing MSP filaments beneath the plasma membrane to generate protrusions in *Ascaris suum* sperm [46]. PP2A dephosphorylates MSP fiber protein 3 (MFP3) to disassemble MSP filaments [37]. PKA may act downstream of UDF, triggering signals to phosphorylate MSP filament-associated proteins and then regulate spermatozoa chemotaxis induced by UDF. However, the molecular mechanism underlying this extracellular ATP-involved sperm chemotaxis requires further elucidation.

Taken together, our work reveals that ATP and PKA signaling pathways are involved in *Ascaris suum* sperm chemotaxis. Future studies on *Ascaris suum* sperm chemotaxis will help us discover inhibitors of the signaling molecules in this process, which can be used to block the reproduction of this harmful parasitic worm. Moreover, the molecular mechanisms involving ATP and PKA signaling pathways in *Ascaris suum* sperm chemotaxis may also exist in mammalian sperm. Further studies on the chemotaxis of nematode sperm will provide insights into mammalian sperm chemotaxis and will be helpful in developing drugs to improve the quality of human sperm.

## 4. Materials and Methods

### 4.1. Preparation and Purification of UDF

Female *Ascaris suum* were collected from slaughterhouse in worm buffer (PBS buffer containing 10 mM NaHCO_3_, pH 7.0) and their uteruses were harvested and ground in HKB buffer (50 mM HEPES, 70 mM KCl, 10 mM NaHCO_3_, pH 7.6). The homogenate was centrifuged at 12,000 rpm for 10 min at 4 °C. Supernatant was heated at 100 °C for 10 min and then centrifuged at 12,000 rpm for 10 min. After removing precipitate, supernatant was centrifuged at 38,000 rpm for 1 h at 4 °C. 100% (NH_4_)_2_SO_4_ concentration was used to extract protein from supernatant. The suspension was centrifuged at 15,000 rpm for 1 h at 4 °C. (NH_4_)_2_SO_4_ precipitate fraction was suspended in HKB buffer for further purification. This fraction was called uterus-derived factor (UDF).

UDF was further fractioned on HiTrap Heparin HP column (GE Healthcare, Madison, WI, USA, 5 mL) equilibrated with HKB buffer and eluted with HKB buffer containing 1.5 M NaCl. The flow through was loaded onto HiTrap Q HP column (GE Healthcare, Madison, WI, USA, 5 mL) equilibrated with HKB buffer and then eluted with a 120 mL linear gradient of HKB buffer containing 0–1.5 M NaCl. The fraction which attracted spermatozoa in elute was further fractioned on a Superdex 200 column (GE Healthcare, Madison, WI, USA, 120 mL) equilibrated with HKB buffer. The fraction which exhibited sperm-attracted activity was concentrated with Amicon Ultra-4 centrifugal filter devices (10 K cutoff) for Alexa Fluor 555 labeling.

### 4.2. UDF Binding Assays

The concentrated Superdex 200 fraction of UDF (2 mg/mL) was labeled with an Alexa Fluor 555 protein labeling Kit (Molecular Probes) according to the manufacturer’s instructions. Spermatids and spermatozoa were incubated with Alexa 555-UDF at 37 °C for 20 min and then washed with HKB buffer. Samples were examined with an LSM710 confocal microscope (Carl Zeiss Microscopy, Oberkochen, Germany). The fluorescence intensities of samples were also measured with flowcytometry. In competition assay spermatozoa were incubated with unlabeled UDF at 37 °C for 20 min and then centrifuged at 2000 rpm for 1 min. The supernatant was removed and cells were rinsed in HKB buffer (pH 7.1) and then incubated with Alexa 555-UDF at 37 °C for 20 min. Samples were observed with an LSM710 confocal microscope. Images were captured with a 63× oil immersion objective and λex 543 nm. The emission fluorescence was filtered at 565 nm. The fluorescence intensities of samples were also measured with flowcytometry.

### 4.3. Transwell Chemotaxis Assays

Spermatids were activated by vas deferens extract (VDE) (1:100 dilution) at 37 °C for 10 min [47]. Transwell assay was performed with Nunc Polycarbonate Membrane (containing 8 µm diameter pores) Inserts in Multidishes. UDF was added in lower wells along the wall and spermatozoa were added in upper wells. The assay device was incubated at 37 °C for 20 min and then upper wells were removed. Spermatozoa in lower wells were counted directly under an Axio Imager M2 microscope (Carl Zeiss Microscopy, Oberkochen, Germany).

### 4.4. Intracellular and Extracellular ATP Measurements

The concentrations of ATP in extracellular media (extracellular ATP) and ATP in cytoplasm (intracellular ATP) were measured with an ATP assay kit (Beyotime, Shanghai, China). Spermatids were washed and then activated by VDE. Spermatozoa (2 × 10^6^ in 500 µL HKB buffer) were incubated with or without 0.5 µg/µL UDF for 20 min. Extracellular media (extracellular ATP) and cell pellet (intracellular ATP) were collected for measuring ATP. Supernatants were collected by centrifugation at 13,000 rpm for 10 min at 4 °C. Precipitated cells were lysed with lysis buffer in ATP assay kit. A 20 µL sample was added into 100 µL luciferase–luciferin buffer mixture. A Sirius luminometer (Berthold Detection Systems) was used to measure luminescence.

### 4.5. Quinacrine and FM 1-43 Labeling

Spermatozoa were loaded with 5 µM quinacrine dihydrochloride (Sigma-Aldrich Company LLC, Saint Louis, MO, USA) for 15 min at 37 °C. Samples were rinsed with HKB buffer and then observed with an LSM710 confocal microscope (Carl Zeiss). For confocal microscopy observation, images were captured with a 63× oil immersion objective. The wavelength of excitation laser was 488 nm, and the emission fluorescence was filtered at 505–530 nm. Spermatozoa were incubated with or without UDF for 20 min at 37 °C and then labeled with 5 µM quinacrine. Fluorescence intensity was analyzed with flow cytometry.

Pretreated sperm were loaded with 5 μg/mL FM 1-43 (Molecular Probes) for 2 min at 37 °C and then washed with HKB buffer (pH 7.1) once. Sperm were observed on a confocal microscope with a 40× oil immersion objective (Leica, Tokio, Japan). Images were obtained with λex 488 nm and λem 505 nm.

### 4.6. Mitochondrial Membrane Potential Measurement

Mitochondrial membrane potential of spermatids and spermatozoa was measured with a JC-1 kit (Beyotime) following manufacturer’s instruction. Spermatids and spermatozoa were stained with JC-1 (5 μg/mL) for 20 min at 37 °C. After being washed twice with washing buffer, spermatids and spermatozoa were analyzed with flow cytometry, and then λex 488 nm and λem 530 were applied to measure JC-1 monomer while λex 525 nm and λem 590 nm were applied for JC-1 aggregates in parallel. The ratio of JC-1 monomer and aggregates indicated mitochondrial membrane potential.

### 4.7. Immunoblot

Immunoblot assay was performed following standard procedures. For detecting serpin secretion during sperm activation with immunoblot assay, anti-As_SRP-1 polyclonal antibody was diluted at 1:10,000. Anti-phosphoserine monoclonal antibody (anti-pS, Millipore) for detecting protein serine phosphorylation level was diluted at 1:2000 and anti-phosphotyrosine monoclonal antibody (anti-pY, Millipore) for detecting protein tyrosine phosphorylation level was diluted at 1:5000.

## Figures and Tables

**Figure 1 ijms-23-04069-f001:**
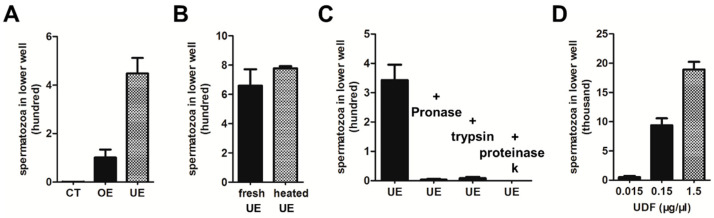
Characterization of uterus-derived factor (UDF). (**A**) Extracts of uterus and oocytes showed activity of attracting spermatozoa in Transwell assay. Activity of uterus extract was stronger than that of oocyte extract. (**B**) Chemoattractant activities of fresh uterus extract and heated uterus extract were similar. (**C**) Activity of uterus extract (UE) was sensitive to proteinase. UE treated by Pronase, trypsin or Proteinase K almost lost the ability of attracting spermatozoa. (**D**) Activity of uterus-derived factor (UDF) in attracting spermatozoa was dose-dependent. Each of these chemotaxis assays were performed three times.

**Figure 2 ijms-23-04069-f002:**
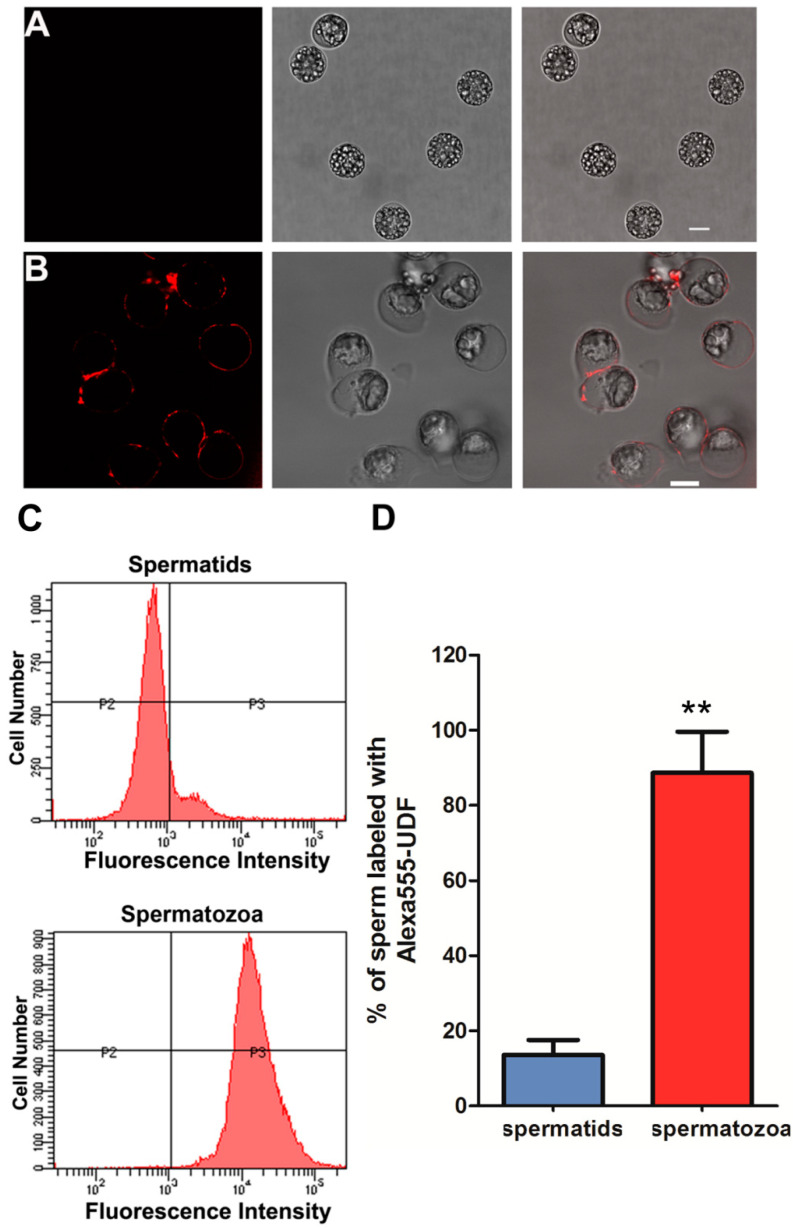
UDF binds spermatozoa but not spermatids. (**A**) Purified UDF was labeled by Alexa Fluor 555 dye. Alexa 555-UDF did not bind spermatids. (**B**) Alexa 555-UDF bound on surface of spermatozoa. (**C**) Number of sperm labeled by Alexa 555-UDF was quantified with flow cytometry assay. Spermatozoa (88%) and spermatids (12%) were labeled by Alexa 555-UDF. (**D**) *X*-axis stands for fluorescence intensity and *y*-axis stands for sperm number in flow cytometry assay. Spermatozoa bound more Alexa 555-UDF than spermatids. (Student’s *t*-test, ** *p* < 0.01). This UDF competition assay was performed twice.

**Figure 3 ijms-23-04069-f003:**
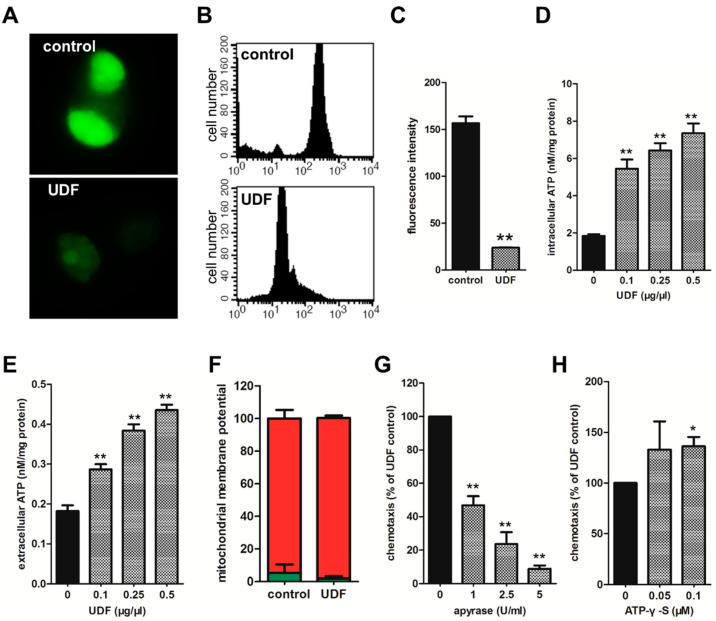
UDF enhances ATP release from cytoplasm to extracellular space in spermatozoa. (**A**) Control spermatozoa and UDF-induced spermatozoa were stained with quinacrine (5 µM) which is an ATP storage marker. (**B**) Fluorescence intensity of quinacrine-labeled spermatozoa which were incubated with or without UDF was measured with flow cytometry. *x*-axis stands for fluorescence intensity and *y*-axis stands for sperm number. (**C**) Fluorescence intensity of quinacrine in spermatozoa induced by UDF declined by 6-fold compared with control. The results were based on three independent replicates. (**D**) Intracellular ATP level of spermatozoa induced by UDF was measured. UDF elevated cytosolic ATP concentration in spermatozoa and the process was dose-dependent. (**E**) UDF elevated extracellular ATP concentration. The process was dose-dependent. UDF treatment enhanced ATP release from cytoplasm to extracellular space. (**F**) Mitochondrial membrane potential was not affected by UDF (0.5 µg/µL). (**G**) Sperm chemotaxis was attenuated when ATP was degraded by apyrase (1 U/mL, 2.5 U/mL and 5 U/mL). (**H**) ATP-γ-S, an analogue of ATP, was able to enhance spermatozoa chemotaxis (Student’s *t*-test, * *p* < 0.05; ** *p* < 0.01). The data shown in (**D**,**E**,**G**,**H**) were based on three independent replicates. Measurements of mitochondrial membrane potential shown in (**F**) were repeated twice.

**Figure 4 ijms-23-04069-f004:**
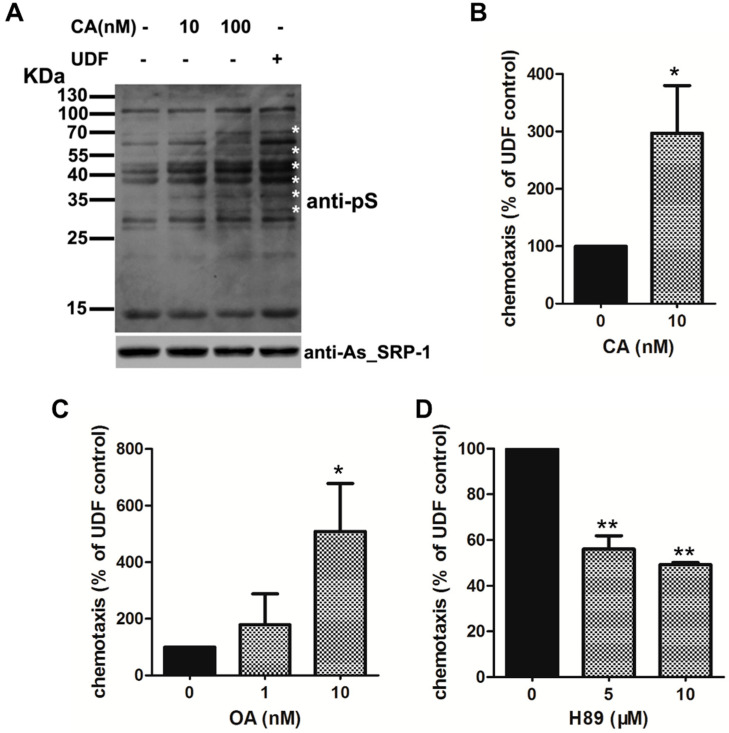
Increasing serine phosphorylation favors spermatozoa chemotaxis promoted by UDF. (**A**) UDF enhanced protein serine phosphorylation level of spermatozoa. Calyculin A (CA, 10 nM and 100 nM), a PP2A/PP2B inhibitor, promoted protein serine phosphorylation. UDF (0.5 µg/µL) also promoted serine phosphorylation. Protein (5 µg) was loaded. Asterisk * represents PP2A substrates whose serine phosphorylation was enhanced. (**B**) CA (10 nM) enhanced spermatozoa chemotaxis promoted by UDF. (**C**) Okadaic acid (OA, 1 nM and 10 nM), an inhibitor of PP2A, favored spermatozoa chemotaxis promoted by UDF. (**D**) H89 (5 µM and 10 µM), an inhibitor of PKA, attenuated spermatozoa chemotaxis. (Student’s *t*-test, * *p* < 0.05; ** *p* < 0.01). The results shown are based on three independent replicates.

## Data Availability

The data underlying this article will be shared upon reasonable request to the corresponding author.

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
