# Peer review of "Released ATP Mediates Spermatozoa Chemotaxis Promoted by Uterus-Derived Factor (UDF) in *Ascaris* *suum"

_ijms, 2022, doi:10.3390/ijms23074069_

Round 1
Reviewer 1 Report
The manuscript is very interesting related to the spermtozoa chemotaxis by uterus-derived factor. Several points could be improved before publication especially in the section of Discussion.
- The authors mentioned that "more than 10 proteins were detected in our final acdtive fractions and failed to identify potential candidate proteins" in the Resultss. It may be better to provide more information and discuss about the potential candidate protein of this species and also try to compare to other known information in the section of Discussion.
- How many replicates or sample number in the results of figures.
- Could you comment and compare the concentration of ATP-S (the ATP analog) with stored or cytosol or extracellular ATP concentration. Why there is no dose-response for Fig 3H?
- Line 233: "but UDF did not affect the mitochnondrial activity of spermatozoa"? based on what?
Reviewer 2 Report
Very interesting manuscript with good results.
The manuscript lacks the explanantion of usefulness and practical use of these results.
Methods are in very good level, interesting results are well discused. Conclusion corresponds to presented results.
minor point:
line44 error point ...egg. (10).
line 192 missing space before (Student....
line 242 superscript in ref. 28
line 293 double point in the end of the line
